# Breast Fluctuating Asymmetry in Women with Macromastia/Gigantomastia

**DOI:** 10.3390/ijerph192416895

**Published:** 2022-12-16

**Authors:** Anna Kasielska-Trojan, Tomasz Zawadzki, Bogusław Antoszewski

**Affiliations:** Plastic, Reconstructive and Aesthetic Surgery Clinic, Medical University of Lodz, Kopcinskiego 22, 90-153 Lodz, Poland

**Keywords:** breast size, fluctuating asymmetry, macromastia

## Abstract

Background: A number of studies have reported breasts have high fluctuating asymmetry (FA:|Right-Left|), suggesting that they contain evolutionary and clinical information (e.g., high FA correlates with breast cancer risk). Here we focus on breast FA in women with a wide range of breast sizes, including participants with macromastia and/or gigantomastia. Material and methods: The sample included 65 women (mean age 33.97 ± 12.1 years). Thirty were randomly selected students and/or patients who regarded their breast size as small, normal or average and who had not undergone or intended to have any breast surgery. The remainder (*n* = 35) were qualified for breast reduction due to macromastia and/or gigantomastia. In all participants we measured/calculated weight, height and BMI, as well as took chest photographs. Breast volumes and nipple areola complex FAs were evaluated in a specifically-designed software. Results: Breast size significantly positively correlated with breast volume FA in all women (t = 5.17, *p* < 0.0001) and in women with macromastia/gigantomastia (t = 2.32, *p* = 0.027). All nipple location FAs correlated positively with breast size. Conclusions: In women with different breast sizes, breast size correlates with FA calculated from breast volume and nipple location FAs. In women with macromastia and/or gigantomastia, breasts present higher FA than “normal” breasts. This observation may give a rationale for earlier and more frequent prophylactic breast imaging in women with macromastia and/or gigantomastia.

## 1. Introduction

Breasts in humans are highly sexually dimorphic due to their size increase in females. Importantly, breast asymmetry increases with breast size and may contain information regarding women’s phenotypic quality [1]. Moreover, these small, random deviations from perfect bilateral symmetry (fluctuating asymmetry [FA]) are often regarded as a measure of the developmental stability of the individual, which is a useful correlate of phenotypic and genetic quality [2]. Møller et al. (1995) reported that large breasts have higher levels of FA than small breasts, breast FA is a predictor of age-independent fecundity and that breast FA appears to be associated with sexual selection [3].

Macromastia and/or gigantomastia is a rare condition characterized by excessive breast growth. Importantly, physical and psychological symptoms are the major criterion for the diagnosis of macromastia and/or gigantomastia, rather than the volume of excess breast tissue that needs to be removed [4,5,6]. It can be the result of hormonal disturbances and changes due to some disorders, or it may arise from physiological conditions such as pregnancy (gestational gigantomastia (GG), [7,8,9]). Another common type of gigantomastia is juvenile enlargement of the breast [4]. In an earlier study, we reported high waist-to-hip ratio (WHR), and particularly high WHR relative to BMI in women with juvenile gigantomastia [10]. WHR may be considered as a further pointer of female quality. It has been shown to reflect health and reproductive capability of women, and those with low WHRs are judged more attractive and healthier [11,12,13]. There are, however, no studies concerning the link between breast size and asymmetry (FA) in women with gigantomastia. In this context, we may suspect that women with juvenile and/or idiopathic macromastia and/or gigantomastia may present high FA. If so, this is of importance, as breast FA may serve as an important clinical marker of breast cancer risk. Hudson et al. (2020) assessed the associations between automated volumetric estimates of mammographic asymmetry and breast cancers. After analysing 79,731 mammographic images, the authors concluded that breast volume asymmetry was positively associated with the risk of an interval cancer diagnosis [1,14,15,16].

In this report, we aimed to consider breast fluctuating asymmetry (FA)—volume and nipple location FAs (|R-L|)—in women with different breast sizes, including women with macromastia and/or gigantomastia.

## 2. Materials and Methods

### 2.1. Participants and Procedure

This prospective study involved 65 women (mean age 33.97 ± 12.08 years): 35 consecutive participants qualified for breast reduction due to idiopathic or juvenile macromastia and/or gigantomastia in one centre (Plastic, Reconstructive and Aesthetic Surgery Clinic) and 30 randomly selected women who described their breasts as small, normal or average and who had not undergone or intended to undergo any breast surgery (Table 1). All patients qualified for surgical treatment had undergone endocrine examinations and a detailed medical interview. They had also received breast ultrasonography or mammography examination. All procedures were performed because of health indications and were funded by the National Health Fund. To meet qualification criteria, patients provided referrals from neurosurgeon, neurologist or orthopedic surgeon acknowledging cervical or back spine disorder and/or poor posture due to heavy breasts. Other clinical symptoms included skin lesions (ulcerations, inflammatory lesions in inframammary folds, on arms), mastalgia, problems with fitting clothes, limitation of social functioning (e.g., sport activities). Exclusion criteria were any hormonal disturbances or treatment (current or past, excluding contraceptives), current or past obesity (BMI > 30 kg/m^2^), pregnancy-related breast enlargement, general health state disabling operation, any abnormalities in breast imaging, history of breast malignancy. To exclude a possible influence of the phase of the cycle on the breast FAs, all photographs were performed in the follicular phase.

### 2.2. Measurements

All women had chest photographs performed according to the requirements of the software BreastIdea (BI) and its module BreastIdea Volume Estimator (BIVE): frontal view and right and left chest profiles. Before performing photographs, reference points were placed on the patient’s body. BI and BIVE were validated for linear and volumetric measurements [17,18]. The following measurements were performed: right and left breast volumes, sternal notch–nipple distances (sn-nip), nipple–midline (nip-ml) distances and difference in nipples’ levels (delta nip-nip’) (Figure 1). All measurements were performed by one researcher. Breast asymmetries were recorded as right minus left breast measurements: volumes (delta V), sternal notch–nipple distances (delta sn-nip), nipple–midline (delta nip-ml) and differences in nipples’ levels (delta nip-nip’). Signs were removed and the asymmetries were analysed as their absolute values (|R-L|). Breast size was calculated as the mean volume of the right and the left breast. In the comparison of breast parameters in women with normal breasts with women with macromastia and/or gigantomastia, breast volume asymmetry was shown as the percentage of delta V in relation to the breast size (mean breast volume).

Additionally, participants’ weight and height were measured, and their BMI was calculated. Direct measurements were made with GPM anthropological instruments (anthropometer). Body height was measured in standing position with the head positioned in the Frankfurt plane.

We randomly selected 10 women from the macromastia group and 10 from those with “normal” breasts and performed all measurements twice to calculate the technical error of measurement (TEM) and the coefficient of reliability (R).

### 2.3. Statistical Methods

An analysis was conducted regarding correlations between breast sizes (mean volumes) and the asymmetries. Furthermore, asymmetry differences between the groups (women with macromastia and those with “normal” breasts) were compared. The asymmetry was analysed as unsigned absolute values of the differences between right and left sides, which implied “half-normal” distribution of the data. The values were log-transformed. Kolmogorov–Smirnov tests showed that that the distribution of the log-transformed variables were normal in all groups. Therefore, the breast asymmetries showed ideal FA (Figure 2 and Figure 3). Due to the possible age influence on breast asymmetry, further analyses included age and breast size (the mean volume of the right and the left breasts) as independent variables and measures of FAs as dependent variables. In case of two variables (delta sn-nip, delta nip-ml) due to perfect symmetry of breasts in some cases, before log transforming unsigned FA’s, “1” was added (log 1 + x). All statistical analyses were performed using the STATISTICA PL package.

## 3. Results

### 3.1. Reliability of Measurements

The coefficient of reliability for raw measurements (R) ranged from 0.986 to 0.999. The highest reliabilities were observed for the left (TEM = 5.81, R = 0.999) and right breast volume (TEM = 7.554, R = 0.999). These values suggest that the observed variability of the ratios is largely due to individual differences rather than measurement error.

### 3.2. Breast Size and Volume Asymmetry

Multiple regression analysis included age and breast size (the mean volume of the right and the left breasts) as independent variables and measures of FAs as dependent variables. The linear regression model for breast volume FA in all women for age and breast size as independent variables (F = 20.45, *p* < 0.0001) and *t* tests showed that age did not correlate with breast volume FA (t = 0.13, *p* = 0.9), while breast size significantly correlated with breast volume FA (t = 5.17, *p* < 0.0001). An increase of 10 mL in mean breast size increases breast volume FA 1.02 times. This model explains 40% of the variance in volume FA (coefficient of determination R^2^ = 0.398). To remove the influence of age and determine a separate influence of breast size, a partial correlation coefficient for this variable was calculated: r = 0.549. Breast size independently explains 26% of the variation in breast volume FA (Table 2).

In women with normal-sized breasts, the multiple regression model for age and breast size and *t* tests detected that age significantly correlates with breast volume asymmetry (t = −2.28, *p* = 0.03), while breast size does not (t = 1.52, *p* = 0.14). When age increases by 1 year, the breast volume FA decreases 0.96 times. This model explains 20% of the variance in volume FA (coefficient of determination R^2^ = 0.2) with a partial correlation coefficient for this variable: r = 0.4. Age independently explains 15% of the variation in breast volume FA (Table 2).

Multifactorial model and *t* tests for age and breast size as independent variables and volume FA as a dependent variable in women with macromastia and/or gigantomastia (F = 5.08, *p* < 0.0122) showed that age does not correlate with breast volume FA in this group (t = 1.08, *p* = 0.29), while breast size significantly correlates with breast volume FA (t = 2.32, *p* = 0.027). An increase of 10 mL in mean breast size increases breast volume FA 1.016 times. This model explains 24% of the variance in volume FA (coefficient of determination R^2^ = 0.241) with a partial correlation coefficient for this variable r = 0.38. Breast size independently explains only 13% of the variation in breast volume FA (Table 2).

### 3.3. Breast Size and Nipple Position Asymmetry

In women with different breast sizes, all nipple FAs correlated with breast size but not with women’s age. In “normal” breasts, only differences in nipple levels correlated positively with breast size. There were no correlations between nipple position FAs and age and breast size in women with macromastia and/or gigantomastia in multifactorial analysis (Table 2).

### 3.4. Breast Asymmetry in Women with Macromastia vs. Women with “Normal” Breasts

Breast volume and nipple position FAs were higher in women with macromastia/ gigantomastia than in those with normal breasts. However, the correlation between volume FA and mean breast volume did not differ between the groups (Table 3).

### 3.5. Breast Size and FA and BMI

BMI appeared to correlate with breast size (the mean breast volume) in all women as well as in women with normal breasts and with macromastia and/or gigantomastia (BMI vs. breast size: all women r = 0.63, women with normal breasts r = 0.41, women with macromastia r = 0.49. Correlation coefficient (r) with BMI was always higher for breast size than breast FAs. Higher BMI increased breasts’ volume, but not their FAs (volume and nipple position).

## 4. Discussion

This study aimed to consider breast FA in women with different breast sizes, including women with macromastia and/or gigantomastia, and discuss how breast FA may indicate their phenotypic quality. The rationale for such speculations was based on studies that showed positive correlations between breast size and asymmetry and breast cancer risk. Kusano et al. (2006) reported that for lean women, larger breast size was associated with a higher risk of breast cancer, while Hudson et al. (2020) presented positive correlations between breast asymmetry and breast cancer diagnosis [14,19]. Scutt et al. (1997, 2006) considered mammograms of women with breast cancer and compared them with healthy controls. The authors found that breast cancer patients had more breast asymmetry and larger breasts than healthy women. They concluded that high breast asymmetry may be a risk factor for breast cancer, while their further study showed that breast asymmetry was greater in healthy women who later developed breast cancer [15,16]. There are, however, no studies available in the literature concerning breast FAs in women with macromastia and/or gigantomastia nor studies on their risk of breast cancer.

In this report, we considered women with a large range of breast sizes (including women with macromastia and/or gigantomastia) and found that volume and nipple location FAs correlate positively with breast size (mean breast volume). Taking into consideration the mentioned studies concerning breast asymmetry and the increased risk of breast cancer, our results may be tentatively interpreted in the clinical aspect. They may influence clinicians’ attention to focus on women with macromastia and/or gigantomastia in regard to more frequent prophylactic examinations, e.g., using ultrasound imaging.

Female breast size may also be an interesting issue regarding evolutionary biology, as breasts may contain information associated with health and fertility. Breasts develop rapidly as a response to the production of sex steroids, mainly oestrogen, which may also act as an immunosuppressant [20,21]. Individuals with a high mutational load and/or those that are homozygous at many loci may show low tolerance to immunosuppression, so large breasts may be a signal of genetic quality. It was also found that breast size increases with body size [1]. We replicated this finding for all women (with different breast sizes), for women with “normal” breasts and for women with macromastia and/or gigantomastia. Higher BMI increased breasts’ volume but not their FAs (volume and nipple position). Previously reported “male” WHR in women with gigantomastia suggests that this condition may be associated with high levels of testosterone and low levels of oestrogen and may not manifest female high phenotypic quality [10]. We also found correlation between breast size (mean breast volume) and volume FA in women with different breast sizes (all women) and in women with macromastia and/or gigantomastia analysed separately. A similar correlation was observed by Manning et al. (1997) for women with “normal” breasts. However, we observed that the effect size of this correlation for women with pathologically large breasts was smaller. This may be related to negative allometry, which may result from the influence of sexual selection. Moreover, we found that volume and nipple location FAs were significantly higher in women with macromastia and/or gigantomastia than in women with “normal” breasts. However, volume FA in relation to mean breast volume did not differ between the groups, which is difficult to interpret and should be considered in further studies.

The present study has some limitations. Although the number of cases may seem small, macromastia and/or gigantomastia are rare conditions. Moreover, we considered only White women, so the results may not be generalizable and should be confirmed on larger groups, e.g., in multicentre studies. Further, women with “normal” breasts differed from women with macromastia and/or gigantomastia in the aspect of BMI and age; however, these effects were removed as BMI did not correlate with breast FA, similar to age, which weakly correlated with volume FA only in women with “normal” breasts.

To conclude, the study suggests that in women with different breast sizes (from “small” and “normal” to macromastia and/or gigantomastia), breast size correlates with their volume and nipple location FAs. Moreover, women with macromastia and/or gigantomastia breasts present higher FA than “normal” breasts. Based on the study by Hudson et al. (2020), this observation may give a rationale for earlier and more frequent prophylactic breast imaging in women with macromastia and/or gigantomastia [14]. Moreover, in the aspect of analysing breast size as a signalling trait, it seems that macromastia and/or gigantomastia may be a signal of low phenotypic quality. Further multicentre studies should be performed to verify this preliminary observation.

## Figures and Tables

**Figure 1 ijerph-19-16895-f001:**
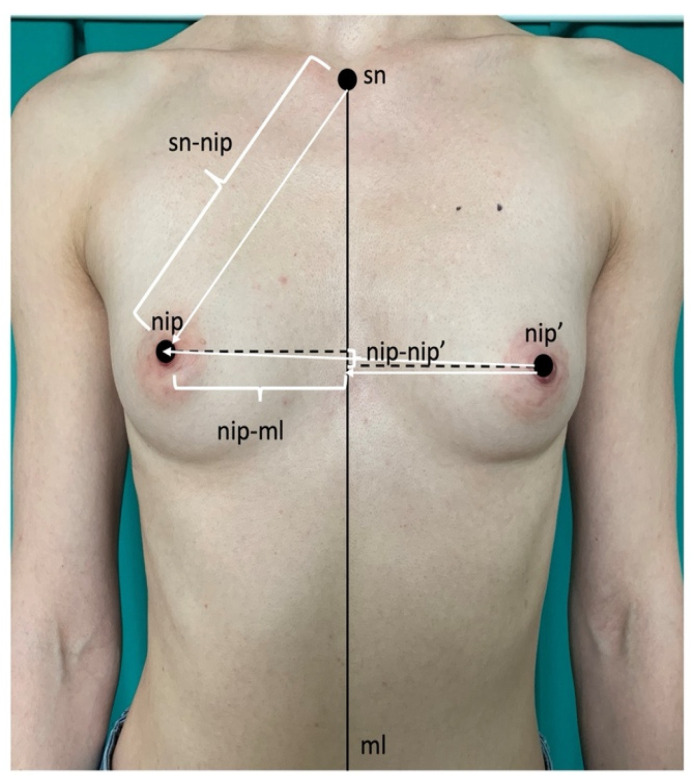
Anthropometric points and measurements analysed in the study.

**Figure 2 ijerph-19-16895-f002:**
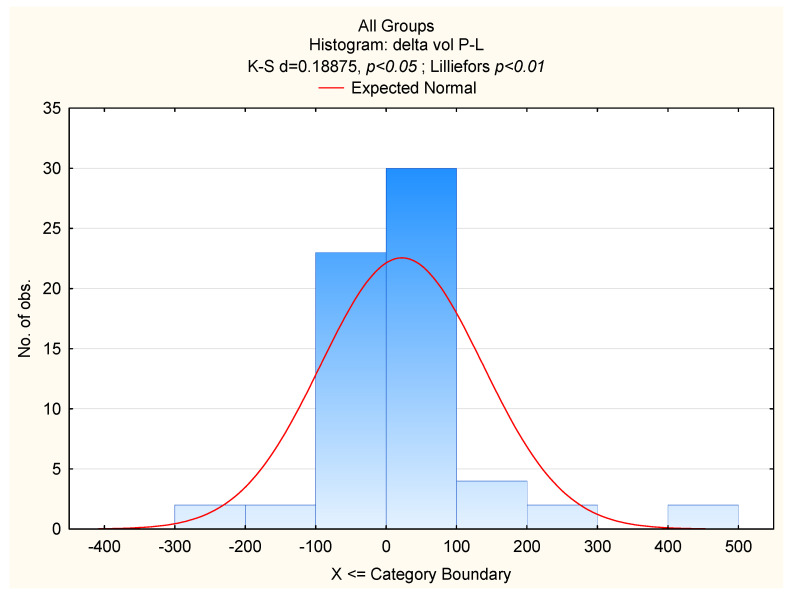
Breast volume fluctuating asymmetry in women with different breast sizes.

**Figure 3 ijerph-19-16895-f003:**
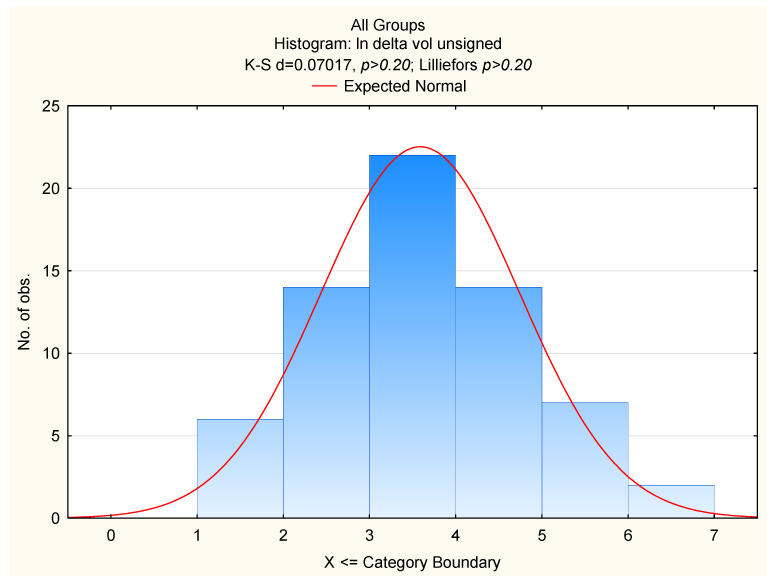
Log of breast volume fluctuating asymmetry in women with different breast sizes.

**Table 1 ijerph-19-16895-t001:** Characteristics of the examined women.

	All Women *n* = 65	Normal Breasts *n* = 30	Macromastia/ Gigantomastia *n* = 35	t/Mann–Whitney	*p*
Age [years]	33.97 ± 12.08	27.53 ± 7.72	39.48 ± 12.49	4.01 *	**<0.0001**
BMI [kg/m^2^]	23.87 ± 2.72	22.45 ± 2.13	25.09 ± 2.60	4.43	**<0.0001**
Breast size—volume [cc]	608.88 ± 377.46	283.08 ± 143.73	888.14 ± 276.82	11.28	**<0.0001**

* *t* test, bold font—t/Mann Whitney test significance.

**Table 2 ijerph-19-16895-t002:** Linear regression models and tests for age and breast size as independent variables and breast volume and nipple position FAs as dependent variables.

	All Women *n* = 65				Normal Breasts *n* = 30				Macromastia/Gigantomastia *n* = 35			
FA	F, *p*	x Age t, *p*	x Size ^1^ t, *p*	R^2^	F, *p*	x Age t, *p*	x Size ^1^ t, *p*	R^2^	F, *p*	x Age t, *p*	x Size ^1^ t, *p*	R^2^
delta V	20.45, <0.001	0.13, 0.9	5.17, **<0.001**	0.4	3.43, 0.047	−2.28, **0.03**	1.52, 0.14	0.2	5.08, 0.012	1.08, 0.29	2.32, **0.027**	0.24
delta nip-nip’	9.08, <0.001	−1.13, 0.26	4.02, **<0.001**	0.23	5.82, 0.021	-	2.41, **0.021**	0.15	3.14, 0.087	-	-	-
delta nip-ml *	2.89, 0.06	-	2.42, **0.02** **	0.08	0.17, 0.84	-	-	-	0.16, 0.85	-	-	-
delta sn-nip *	12.56, <0.001	0.41, 0.68	3.86, **0.0003**	0.29	0.54, 0.59	-	-	-	2.77, 0.08	-	2.31, **0.03** **	0.14

R^2^—coefficient of determination, bold font—*t* test significance. ^1^—breast size as a mean volume of right and left breast. * FA analysed as (log (1 + unsigned FA) **—after excluding age from the multifactorial analysis.

**Table 3 ijerph-19-16895-t003:** Comparison between FAs of normal breasts versus macromastia/gigantomastia.

FA	Normal Breasts *n* = 30				Macromastia/ Gigantomastia *n* = 35				Mann–Whitney Z	*p*
	Mean	SD	Median	Q1; Q3	Mean	SD	Median	Q1; Q3		
delta V [cc]	24.83	19.72	20.56	10.84; 31.77	108.4	113.44	71.55	24.91; 160.25	4.283	**<0.0001**
delta V/mean V [%]	10.1	7.02	9.26	3.68; 14.68	11.36	9.53	9.26	3.50; 16.48	0.217	0.828
delta nip-nip’ [cm]	0.7	0.46	0.57	0.37; 1.02	1.79	1.54	1.61	0.42; 2.55	2.967	**0.003**
delta sn-nip [cm]	0.51	0.37	0.47	0.18; 0.86	1.55	1.40	1.17	0.40; 2.53	3.323	**0.001**
delta nip-ml [cm]	0.69	0.57	0.60	0.34; 0.93	1.08	0.70	1.01	0.54; 1.47	2.316	**0.021**

Bold font—test significance.

## Data Availability

Data available on request from the corresponding author.

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
