# Peer review of "Breast Fluctuating Asymmetry in Women with Macromastia/Gigantomastia"

_ijerph, 2022, doi:10.3390/ijerph192416895_

Round 1
Reviewer 1 Report
Thank you for the opportunity to review this interesting article. The authors present an analysis of the fluctuating asymmetry, breast size and macromastia/gigantomastia. The paper is easy to read and well written. I recommend its publication after introducing some minor remarks:
1. Could you explain in point 2.1 Participants and Procedure what were the health indications for breast reduction in described group of patients qualified for such procedure?
2. I suggest to add a simple diagram or even a photograph of patient on which you place all references points needed to perform described measurements. It could be helpful for readers and researchers.
3. The discussion is very well described, I encourage the authors to continue their research as the obtained conclusions are very interesting.
Author Response
Dear Editor and Reviewers,
Thank you for your interest in our manuscript entitled " Fluctuating asymmetry, breast size and macromastia/gigantomastia”. We would like to thank anonymous Reviewers for their detailed valuable comments and reviews, which helped to improve the manuscript. In this revision we addressed all your comments (in the text changes are marked in red). We hope that this revision meets with your approval.
Sincerely,
The Authors
Responses to Reviewer’s comments:
Reviewer 1
- Thank you for the opportunity to review this interesting article. The authors present an analysis of the fluctuating asymmetry, breast size and macromastia/gigantomastia. The paper is easy to read and well written. I recommend its publication after introducing some minor remarks:
Thank you for your recommendation!
- Could you explain in point 2.1 Participants and Procedure what were the health indications for breast reduction in described group of patients qualified for such procedure?
As suggested such information was included:
“To meet qualification criteria patients provided referrals from neurosurgeon, neurologist or orthopedic surgeon acknowledging cervical or back spine disorder and/or poor posture due to heavy breasts. Other clinical symptoms included: skin lesions (ulcerations, inflammatory lesions in inframammary folds, on arms), mastalgia, problems with fitting clothes, limitation of social functioning (e.g. sport activities).”
- I suggest to add a simple diagram or even a photograph of patient on which you place all references points needed to perform described measurements. It could be helpful for readers and researchers.
As suggested, a photograph illustrating the performed measurements was added.
- The discussion is very well described, I encourage the authors to continue their research as the obtained conclusions are very interesting.
Thank you!

Reviewer 2 Report
The manuscript describes a study with undefined design (retrospective? prospective? randomized?) based on data collected about breast size and asymmetry of 65 female patients. The authors positively correlate the fluctuating asymmetry (FA) with with macromastia and/or gigantomastia. However, despite an extensive and absolutely remarkable statistic, they failed to provide solid conclusions on the relationship about fluctuating asymmetry and breast cancer risk.
In detail my remarks:
· line 2-3, the title seems vague and somehow ambiguous
· What is the exact definition of macromastia /gigantomastia? How was determined the size of the breast? How was determined in which group enroll each the patient?
· Table 1, in the study group (women with macromastia) patients are medially 12 years older in median respect the control group, how can you exclude that higher risk for breast cancer is related to age?
· line 229-239, the inclusion/exclusion criteria, the study population and the design of this study cannot be compared with the ones of reference articles cited. So, were the correlation and the rationale appropriated?
Ref n#14 Hudson “..which targets females aged 50–70 years (with a trial for 50% of females aged 47–50 and 70–73) and has a coverage of approximately 75%. Also included were small numbers of younger females who had been identified as having a higher risk of breast cancer and therefore invited for screening on an annual basis, plus any females over 73 years who had optionally contacted the service for a self-referred screening appointment.” ““Ethnicity was categorised according to the Census classification29 and summarised as, “Asian” (Indian, Pakistani or Bangladeshi or other), “Black-African,” “Black-British or Caribbean or other,” “Chinese,” “Mixed” (White and Black, White and Asian or any other mixed), “White” (British or Irish or other) and “Other.” Data for other known breast cancer risk factors (e.g., reproductive history, body mass index (BMI), family-history of breast cancer) are not collected in a systematic way across the NHSBSP screening programme and thus were unavailable”
Ref n#15 Scutt “The original study collected detailed breast cancer risk factor data from a total of 12,942 women self-referred to the Liverpool Breast Screening Unit between 1979 and 1986”, “The controls were from the same risk factor study” “The control group was composed of 163 peri- or post-menopausal women, 76 pre-menopausal subjects, and 13 subjects for whom menopausal status was not recorded.”
Ref n#19 Kusano “The Nurses' Health Study II (NHS II) is an ongoing, prospective cohort of 116,671 female registered nurses. NHS II was initiated in 1989 and enrolled women who at that time were aged between 25 and 42 years and living within 14 states in the United States.” “In the 1993 questionnaire, participants were asked “What was the cup size of your bra when you were 20 years old? (Estimate if you did not wear a bra.).” Participants were given the answer options “A or smaller”, “B”, “C” and “D or larger.”
· line 243-244, currently, clinical interpretation is not supported by evidence in the literature.
· line 268-269, volume FA respect mean size is similar in the two groups, may it depend selectively on the glandular or the fatty tissue, whose proportion varies in women with different volume of breast/age/race?
· line 275-278 “Further, women with “normal” breasts differed from women with macromastia and/or gigantomastia in the aspect of BMI and age, however these effects were removed as BMI did not correlate with breast FA, similar to age, which weakly correlated with volume FA only in women with “normal” breasts.” BMI and age do correlate with breast cancer risk by the literature, how do you address this apparent contradiction?
Globally well written and described, the study is fully focused on fluctuating asymmetry in a very small series of patients, the impossibility of a real patient stratification represents the main limitation that preclude any definitive conclusion.
In my opinion, the authors should avoid including in the text any mention of correlated breast risk, thus limiting the discussion on the distribution of FA among the study population.
Author Response
Dear Editor and Reviewers,
Thank you for your interest in our manuscript entitled " Fluctuating asymmetry, breast size and macromastia/gigantomastia”. We would like to thank anonymous Reviewers for their detailed valuable comments and reviews, which helped to improve the manuscript. In this revision we addressed all your comments (in the text changes are marked in red). We hope that this revision meets with your approval.
Sincerely,
The Authors
Responses to Reviewer’s comments:
Reviewer 2
- The manuscript describes a study with undefined design (retrospective? prospective? randomized?) based on data collected about breast size and asymmetry of 65 female patients.
According to the suggestion, the design of the study was specified:
“This prospective study involved 65 women (mean age 33.97±12.08 years): 35 consecutive participants qualified for breast reduction due to idiopathic or juvenile macromastia and/or gigantomastia in one centre (Plastic, Reconstructive and Aesthetic Surgery Clinic) and 30 randomly selected women who described their breasts as small, normal or average and who had not undergone or intended to undergo any breast surgery”
- line 2-3, the title seems vague and somehow ambiguous
The title was changed to focus on the main idea of the ms: “Breast fluctuating asymmetry in women with macromastia/gigantomastia”
- What is the exact definition of macromastia /gigantomastia? How was determined the size of the breast? How was determined in which group enroll each the patient?
The “definition” of gigantomastia and criteria of diagnosis were described in the introduction:
“Macromastia and/or gigantomastia is a rare condition characterized by excessive breast growth. Importantly, physical and psychological symptoms are the major criterion for the diagnosis of macromastia and/or gigantomastia, rather than the volume of excess breast tissue that needs to be removed”. To further explain this – clinical symptoms are now considered the major criteria for the diagnosis, earlier we used criterium od 1500 g resection, now such approach was abandoned.
Criteria of selecting patients to both groups were described in Material and methods:
“All procedures were performed because of health indications and were funded by the National Health Fund. To meet qualification criteria patients provided referrals from neurosurgeon, neurologist or orthopedic surgeon acknowledging cervical or back spine disorder and/or poor posture due to heavy breasts. Other clinical symptoms included: skin lesions (ulcerations, inflammatory lesions in inframammary folds, on arms), mastalgia, problems with fitting clothes, limitation of social functioning (e.g. sport activities).” and “…and 30 randomly selected women who described their breasts as small, normal or average and who had not undergone or intended to undergo any breast surgery”
- Table 1, in the study group (women with macromastia) patients are medially 12 years older in median respect the control group, how can you exclude that higher risk for breast cancer is related to age?
We did not analyzed the cancer risk directly, just discussed breast asymmetry as a risk factor for breast cancer, the results /breast asymmetry/ were controlled for age statistically.
- line 229-239, the inclusion/exclusion criteria, the study population and the design of this study cannot be compared with the ones of reference articles cited. So, were the correlation and the rationale appropriated?
Yes, but the cited studies did not include women with gigantomastia/macromastia, these were populational studies including large number of women with mostly normal-sized breasts. These studies gave a theoritical background for the study of breast FA in women with gigantomastia and enabled discussing the results in “oncological” aspect. Our results cannot be compared with these studies as our target group is completely different and our study sample is much smaller (which is however not surprising when we consider that we analyzed women with a rather rare condition - gigantomastia)
- line 243-244, currently, clinical interpretation is not supported by evidence in the literature.
Yes, we agree with you, the sentence was misleading. Now, it was rephrased so it refers to the previously discussed papers and results, and this was our intention to refer to them:
“Taking into consideration the mentioned studies concerning breast asymmetry and the increased risk of breast cancer, our results may be tentatively interpreted in the clinical aspect. They may influence clinicians’ attention to focus on women with macromastia and/or gigantomastia in regard to more frequent prophylactic examinations, e.g. using ultrasound imaging. “
- line 268-269, volume FA respect mean size is similar in the two groups, may it depend selectively on the glandular or the fatty tissue, whose proportion varies in women with different volume of breast/age/race?
Thank you for this remark – this would be very interesting to determine breast main components /glandular versus fatty tissue/ in relation to breast FA. I do not know such studies. In fact most (but still not all) women with gigantomastia have a lot of fatty tissue in breasts (despite normal BMI).
- line 275-278 “Further, women with “normal” breasts differed from women with macromastia and/or gigantomastia in the aspect of BMI and age, however these effects were removed as BMI did not correlate with breast FA, similar to age, which weakly correlated with volume FA only in women with “normal” breasts.” BMI and age do correlate with breast cancer risk by the literature, how do you address this apparent contradiction?
I do not see any contradiction here, as, again – we did not analyze the risk of breast cancer, but discussed our results in this possible aspect (refereing to previous studies showing breast asymmetry as a risk of breast cancer). Our results refer to normal-weighted women and were controlled for age. Age is a risk factor for most cancers, also breast cancer, but this realtion may not be related to age ralated changes of breast asymmetry.
- Globally well written and described, the study is fully focused on fluctuating asymmetry in a very small series of patients, the impossibility of a real patient stratification represents the main limitation that preclude any definitive conclusion.
We mentioned our sample size as the main limitation (that is a common limitation of clincial studies of rather rare conditions):
“Although the number of cases may seem small, macromastia and/or gigantomastia are rare conditions”
- In my opinion, the authors should avoid including in the text any mention of correlated breast risk, thus limiting the discussion on the distribution of FA among the study population.
We limited interpreting our results in this aspect, we just mentioned this in a speculative way.
